# Cartesian Mesh Generation with Local Refinement for Immersed Boundary Approaches

Luca Di Angelo *, Francesco Duronio, Angelo De Vita and Andrea Di Mascio

Department of Industrial and Information Engineering and Economics, University of L'Aquila,
Piazzale E. Pontieri, Monteluco di Roio, 67100 L'Aquila, Italy; francesco.duronio@graduate.univaq.it (F.D.);
angelo.devita@univaq.it (A.D.V.); andrea.dimascio@univaq.it (A.D.M.)
* Correspondence: luca.diangelo@univaq.it

**Abstract:** In this paper, an efficient and robust Cartesian Mesh Generation with Local Refinement for an Immersed Boundary Approach is proposed, whose key feature is the capability of high Reynolds number simulations by the use of wall function models, bypassing the need for accurate boundary layer discretization. Starting from the discrete manifold model of the object to be analyzed, the proposed model generates Cartesian adaptive grids for a CFD simulation, with minimal user interactions; the most innovative aspect of this approach is that the automatic generation is based on the segmentation of the surfaces enveloping the object to be analyzed. The aim of this paper is to show that this automatic workflow is robust and enables to get quantitative results on geometrically complex configurations such as marine vehicles. To this purpose, the proposed methodology has been applied to the simulation of the flow past a BB2 submarine, discretized by non-uniform grid density. The obtained results are comparable with those obtained by classical body-fitted approaches but with a significant reduction of the time required for the mesh generation.

**Keywords:** Cartesian adaptive grids; immersed boundaries; LES simulation

## 1. Introduction

The increasing popularity of Computational Fluid Dynamics (CFD) in marine engineering sciences, observed in the last few decades, is to be ascribed to the growth of computational power, in combination with the increase of robustness and accuracy of CFD solvers. Today, Reynolds averaged Navier–Stokes simulations on body-fitted meshes are commonly performed in naval architecture, in order to save time in the design process and make it less expensive; conventional towing tank or water channel tests are usually limited to a few shapes obtained in the final design. Nevertheless, the bottleneck of the whole simulation procedure remains mesh generation; in order to obtain a mesh that satisfies the requirements of smoothness and proper clustering, particularly in boundary layers and wakes, its generation still requires lengthy human interaction and relevant expertise by the user.

Nowadays, the most used method for geometry discretization is the body-fitted approach, particularly for high Reynolds number flows, and most solvers handle unstructured or block-structured grids, possibly with partial overlapping: their generation remains the most demanding task in the total effort and time for the complete simulation [1,2]. The major reason for this last aspect is that the process is never completely automatic, except in those cases where the geometry is so simple that it is possible to parametrize its shape. This is even more complicated in optimization algorithms, where only minor model changes in shape (and not in topology) are allowed.

In the two last decades, the Immersed Boundary (IB) method has emerged as a valid alternative to the body-fitted meshes-based CFD methods. The key idea of IB methods is to locally modify the governing equations in order to enforce the boundary conditions without a body-fitted mesh: this avoids the complex and time-consuming body-fitted

meshes generation, by allowing the discretization with a simple structured Cartesian mesh. The most important advantages of this technique are the easy grid generation, also when dealing with moving boundaries. The IB idea can be attributed to Peskin et al. [3], who used it to simulate cardiac mechanics and the associated blood flow. Nowadays, many variants and procedures of the IB original ideas are proposed: in order to enforce the boundary condition on the interfaces, some exploit a continuous forcing term in the field equations, others explicitly locally modify the equations; excellent reviews can be found, for instance, in [4,5]. With the introduction of wall models [6,7], the use of IB methods was applied also to resolve high Reynolds number flows, mitigating the need for accurate resolution within the boundary layer.

This paper proposes an efficient and robust Cartesian Mesh Generation with Local Refinement for Immersed Boundary Approaches. In particular, the proposed methodology was developed to be suited for the method proposed in [8], which couples the Immersed Boundary approach to the level-set method, and also makes use of wall functions at rigid walls. Although several methods have been published in the literature [8], none of them completely satisfies the requirements of the specific IB considered here; furthermore, for the authors' knowledge, none considers the differential geometric properties of the model surfaces to be analyzed to define an optimized geometry-based discretization. The proposed method aims to overcome these limitations by introducing strategies of diversification of the mesh dimensions in the different parts of the model, based on automatic segmentation of the surfaces enveloping the object.

The proposed methodology, i.e., the Cartesian Mesh Generation with Local Refinement and the IB method with wall functions, is applied to study the flow past a submarine at a high Reynolds number. The obtained results show how the use of the proposed CFD tools is extremely helpful to capture the main flow characteristics in the wake of all the appendages, although the details in the boundary layers are lost because of the adoption of wall functions. The reported results suggest that the use of immersed boundary approach is mature enough to be used as an investigation tool in naval architecture.

## 2. The Cartesian Mesh Generation Method

The proposed Cartesian mesh generation method is specifically developed for the IB method developed in [1]; the approach produced a Cartesian grid with the following features:

- a signed distance from the wall (positive inside the body and negative outside) is defined at each point;
- the mesh can be easily refined close to the boundary and where the solution requires finer discretization (typically in the wake);
- it can consist of block structured Cartesian blocks with possible partial overlapping.

Particular attention is given to the data structure, in order to optimize the storage and minimize interfaces, in view of parallel calculation.

In the related literature, many methods have been published for the generation of Cartesian grids (the interested reader can find details in [8]), although none completely suited to the needs of the developed flow solver. Therefore, a specific grid generation algorithm, whose flow-chart is shown in Figure 1, is implemented.

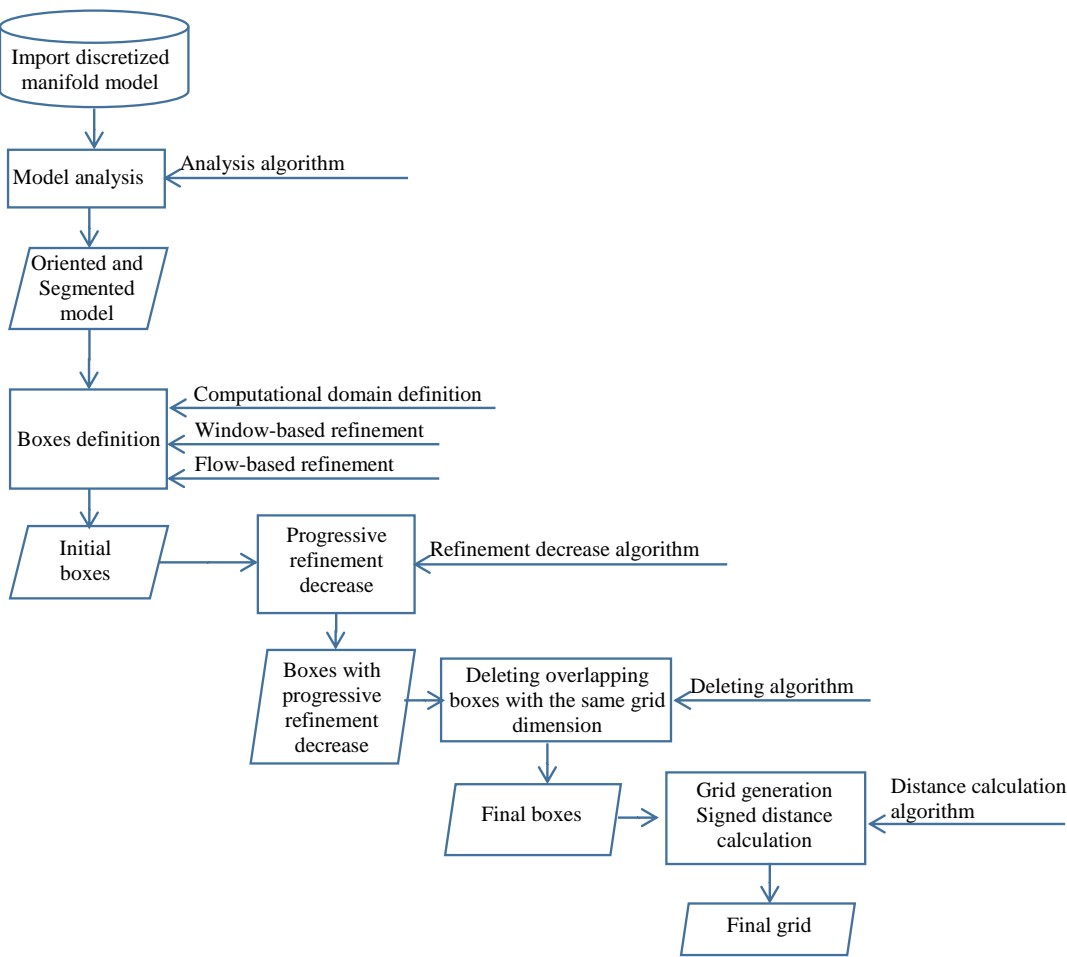

**Figure 1.** Flow-chart of the proposed method.

*2.1. Model Analysis*

The proposed algorithm starts from a discrete manifold model (in STL format) and store it into two tables (named as "Points" and "Triangles") containing the information about the planar triangular facets (Figure 2a):

- **Points** $(x_i, y_i, z_i)$ for $i = i, \ldots, np$: where the coordinates of the $np$ unique points are stored;
- **Triangles**: where three pointers to **Points** are stored for each triangle.

The structure of the two tables avoids redundancy of information. The model is then positioned by rigid roto-translation operations in the Global Reference System $(O, x, y, z)$ of the computational domain so that:

$$\begin{cases} x_{\min}^M = 0 \\ y_G^M = 0 \\ z_G^M = 0 \\ \text{flow direction } // \ x\text{-axis} \end{cases}$$

Here, $\{X_G^M, Y_G^M, Z_G^M\}$ are coordinates of the model centroids where the origin of the new reference frame is translated; all the coordinates of the **Points** are then recomputed in this reference system (Figure 2b). The model processing includes an automatic surfaces segmentation, based on a fuzzy analysis of the discrete differential properties, according to the method proposed in [9] (Figure 2c).

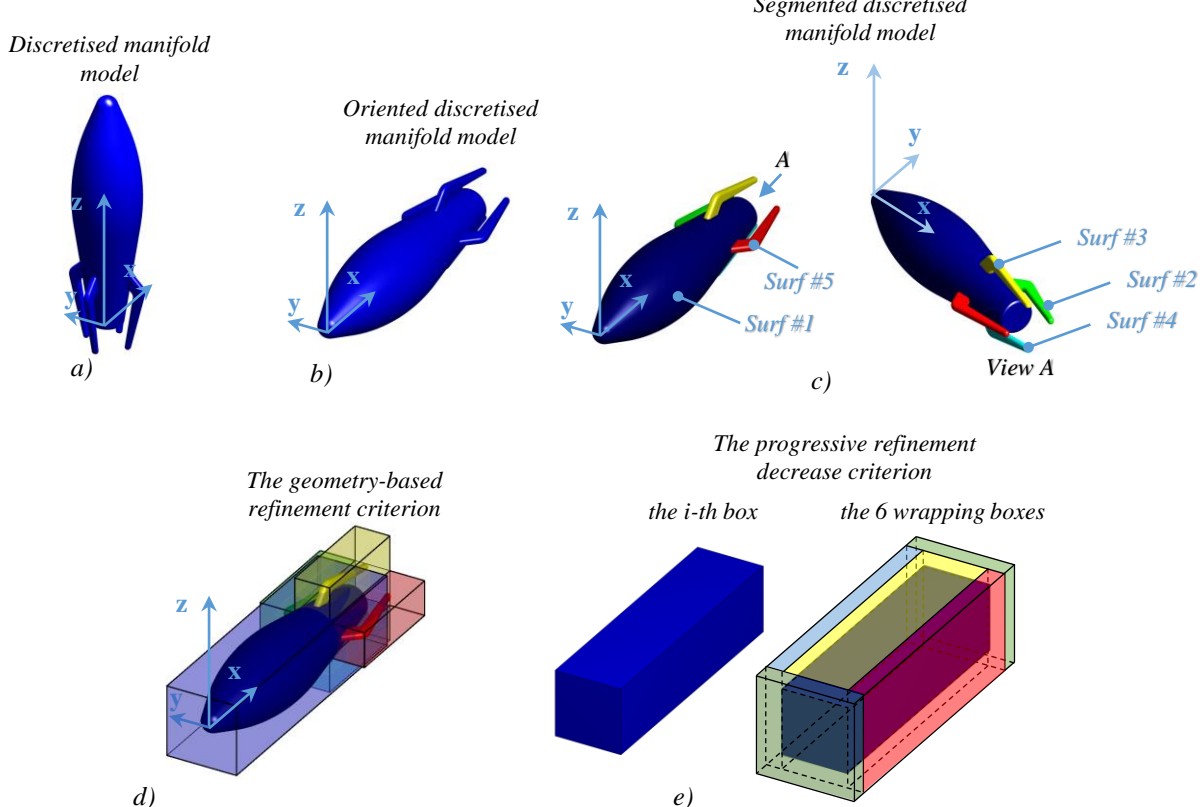

**Figure 2.** Key steps of the proposed grid generation method: (**a**) the imported discretised manifold model; (**b**) the oriented discretised manifold model; (**c**) the results of the discretised manifold model segmentation; (**d**) the boxes generated with the geometry-based refinement criterion; (**e**) the boxes generated with the progressive refinement decrease criterion.

### 2.2. Boxes Definition

The sole explicit operation from the user is the definition of the computational domain where local refinement is required on the basis of the expected flow structure (i.e., wakes); this is done by the definition of a box, identified by the two extreme vertices $\{X_{\min}, Y_{\min}, Z_{\min}\}$ and $\{X_{\max}, Y_{\max}, Z_{\max}\}$ in the Global Reference System and by the size of the far-boundary cells $\left\{\Delta x^{far}, \Delta y^{far}, \Delta z^{far}\right\}$, where $\Delta x^{far} = \Delta y^{far} = \Delta z^{far} = h \times 2^{k_G}$, where $k_G$ is an integer and $h$ is the minimum cell size. The first generated grid is a set of hexahedra having face normal oriented along in the three axes directions of the Global Reference System. The number of Voxels along the three directions is defined as follows:

$$\begin{cases} N_x = \max\left(1, int\left(\frac{X_{\max} - X_{\min}}{\Delta x^{far}}\right)\right) \\\\ N_y = \max\left(1, int\left(\frac{Y_{\max} - Y_{\min}}{\Delta y^{far}}\right)\right) \\\\ N_z = \max\left(1, int\left(\frac{Z_{\max} - Z_{\min}}{\Delta z^{far}}\right)\right) \end{cases} \quad (1)$$

Each of the $N_v = N_x \cdot N_y \cdot N_z$ cells is identified by three sets of generalized indices, defined in the following:

- Equivalent structure cell indices $G_{ijk} = (G_i, G_j, G_h)$ where $1 \leq G_i \leq N_x$, $1 \leq G_j \leq N_y$, $1 \leq G_h \leq N_z$;
- Three coordinates of the center $C_{ijk} = (C_{ijk,x}, C_{ijk,y}, C_{ijk,z})$;
- Refinement level index $K_{ijk} = k_G$.

The complete grid with proper local size is generated by implementing the following refinement criteria, applied to the initial grid:

- geometry-based criterion: refinement based on the distance from the wall surface of the model;
- flow-based criterion: refinement defined on the bases of flow features, in regions with relevant variations of the fluid-dynamic quantities (for example, pressure gradient and vorticity);
- explicit window-based criterion: any other refinement in regions of interest.

The first criterion is automatic; the second and the third ones require that the operator defines each of the $N_{ROI}$ region of interest by its two extreme vertices $\{X_{min}^w, Y_{min}^w, Z_{min}^w\}_l$ and $\{X_{max}^w, Y_{max}^w, Z_{max}^w\}_l$ (where $1 \le l \le N_{ROI}$) coincident with two vertices of the initial grid and the refinement level as $2^{k_W}$ (with the integer $k_W < k_G$). In the sequel of this research activity, the last two will also be rendered fully automatic.

The geometry-based refinement criterion is based on the surfaces segmentation (Figure 2d): for the $i$-th segmented surface, the algorithm:

- creates a box (whose extreme vertices $\{X_{min}^M, Y_{min}^M, Z_{min}^M\}$ and $\{X_{max}^M, Y_{max}^M, Z_{max}^M\}$ coincide with the initial grid) that contains it;
- calculates a value of the grid dimension ($2^{k_{M,i}}$ with the integer $k_M, i < k_G$) on the basis of the surface minimum characteristic dimension.

Regardless of the refinement criteria, the boxes are generated with the extreme points coincident with grid nodes of the computational domain. This allows for keeping the consistency of the discretization schemes also with local refinements.

### 2.3. Progressive Coarsening

To guarantee the smoothness of the refinement level in each direction between a cell and its neighborhoods, an isotropic recursive algorithm working is implemented. At each iteration, six wrapping boxes are generated around each box for which $k_i < \frac{k_G}{2}$ (Figure 2e) with grid dimension $k_{i,new} = 2 * k_i$. The algorithm stops when all the outermost boxes within the computational domain have grid dimension $k_i < \frac{k_G}{2}$.

### 2.4. Removal of Overlapping Boxes with the Same Grid Dimension

The above processes can generate prisms with the same grid dimension that overlaps: to reduce this redundancy of information, a voxelization-based method with subsequent clustering of adjacent prisms is implemented.

### 2.5. Grid Generation with Signed Distance Calculation

For each Voxel of each box, the vertices are generated according to the scheme of Figure 3. In order to minimize redundancy, all this information is stored in two tables:

- ***nodes*** $(x_i, y_i, z_i)$ for $i = 1, \dots, N_n$, where the coordinates of the $N_n$ unique vertices are stored;
- ***Voxels*** where, for each Voxel, the eight pointers to ***nodes*** are stored.

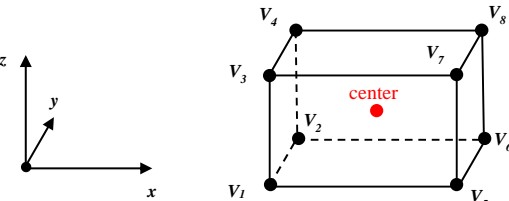

**Figure 3.** The scheme generation of the vertices of each voxel.

The fourth column of the table ***nodes*** is the signed distance between the corresponding node and the model, whose sign encodes whether the point is *inside* (negative) or *outside* (positive) to the watertight surface. The value of the distance is evaluated by searching the minimum distance between each node and some points generated parametrically

on the triangular faces of the model. The distance sign is defined, for its simplicity and computational efficiency, by using a ray-tracing technique [8].

## 3. Case Study

The case study analyzed in this paper is a BB2 submarine with casing and appendage taken from https://www.marin.nl/markets/defence/naval-subsurface-vessel-hydrodynamic-design-services, accessed on the 24 February 2021 (Figure 4). The model has foreplanes on the sail and tailplanes; no propulsion systems and mobile appendages or rudders are present.

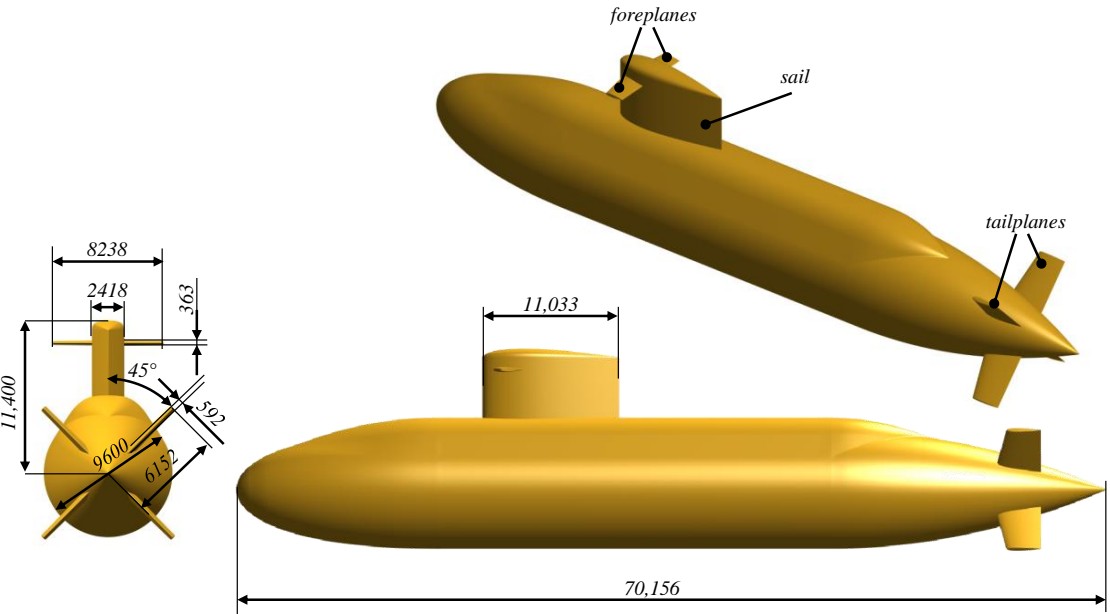

**Figure 4.** The analyzed BB2 submarine with some of the characteristic dimensions.

The choice of this model was done to apply the proposed method to a practical geometry of interest for naval architecture. The Cartesian grid generation of this model is critical because of the different characteristic dimensions and shape of the appendages.

### 3.1. The Refined Cartesian Meshes' Generation

The original solid model was transformed into a discrete model defined by triangular flat faces. Figure 5a shows the oriented model with the bounding box dimension and points' density measures. The clear non-uniformity of points distribution permits to analyze the robustness of the proposed generation method. In Figure 5b, the fundamental characteristics of the computational domain defined for the CFD analysis are depicted. First of all, the far boundaries are placed far enough to minimize blockage effects; then, once grid dimension ($k_G$) is defined, its extreme points ($\{X_{min}, Y_{min}, Z_{min}\}$ and $\{X_{max}, Y_{max}, Z_{max}\}$) were recomputed in order to obtain integer numbers of cells according to (1). Figure 6 shows the results of the surfaces segmentation method superimposing the auto-generated geometry-based boxes refinement. The algorithm correctly recognizes and segments eight surfaces and calculates the $k_{M,i}$ integers on the base of the corresponding surface minimum characteristic dimension.

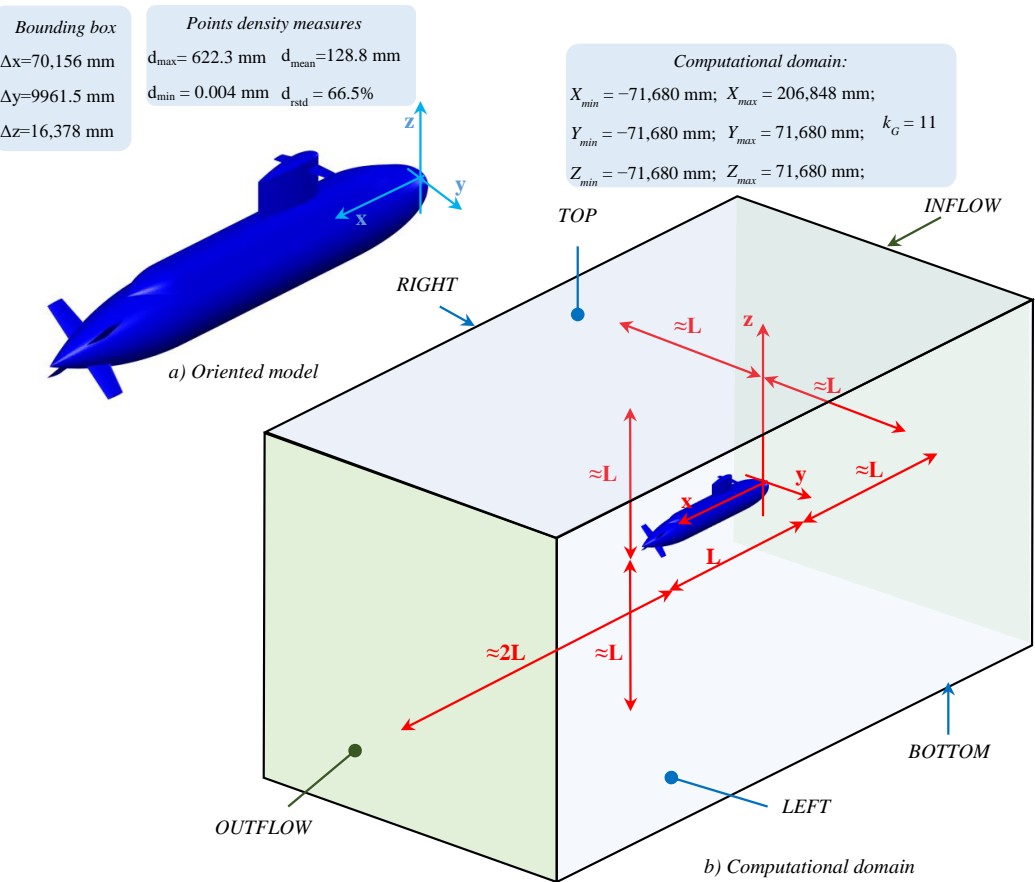

**Figure 5.** The oriented model (*a*) and the computational domain (*b*).

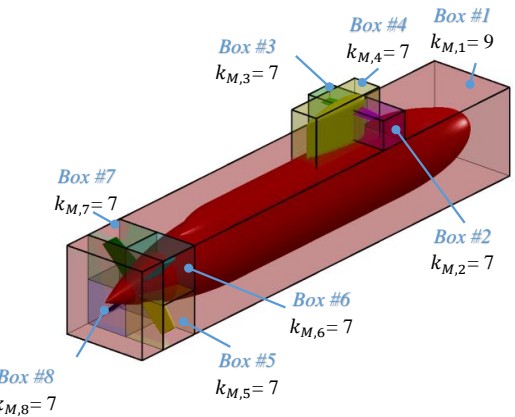

**Figure 6.** The segmented model with geometry-based boxes refinement.

For the application of the proposed algorithm to a realistic CFD analysis, three refinement boxes are introduced to properly capture the expected wakes behind the sail, foreplanes, and tailplanes (Figure 7).

Once the computational domain and the refinement windows are defined, the grid is generated according to the operations shown in Figure 1 and discussed in Section *Grid generation*; for the test case under consideration, the generated grid consists of 12.8 million cells. This value is about 20 times smaller than that used in [10] to analyze the same geometry, simplified by eliminating the two foreplanes. Figure 8a shows the zero level of the signed distance function with superimposing the grid sections on three perpendicular planes, whereas Figure 8b highlights the difference between the original shape and the one obtained by interpolation from the signed distance function. From the figure, it can be seen

that the larger errors are close to the sharp trailing edges of the profiles which, anyhow, are between 60 mm and 100 mm with a maximum error of always <0.14%.

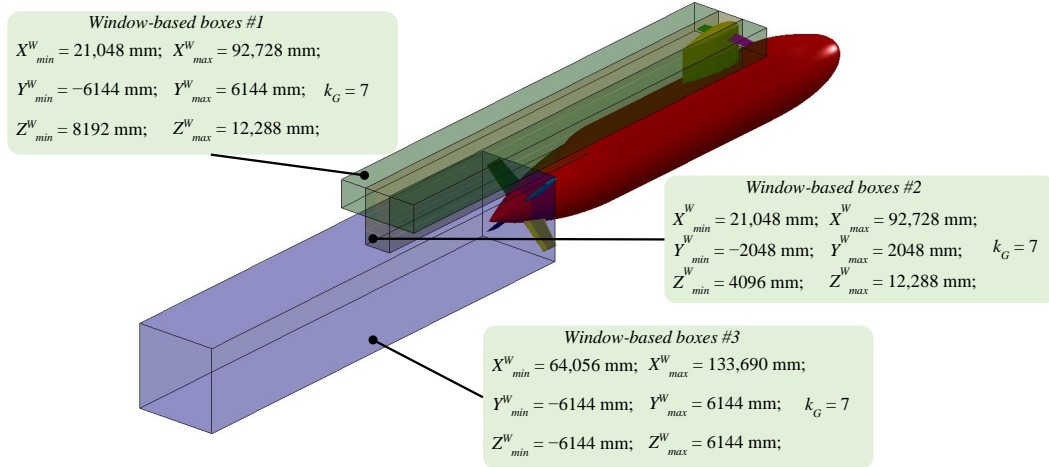

**Figure 7.** The segmented model with window-based boxes refinement.

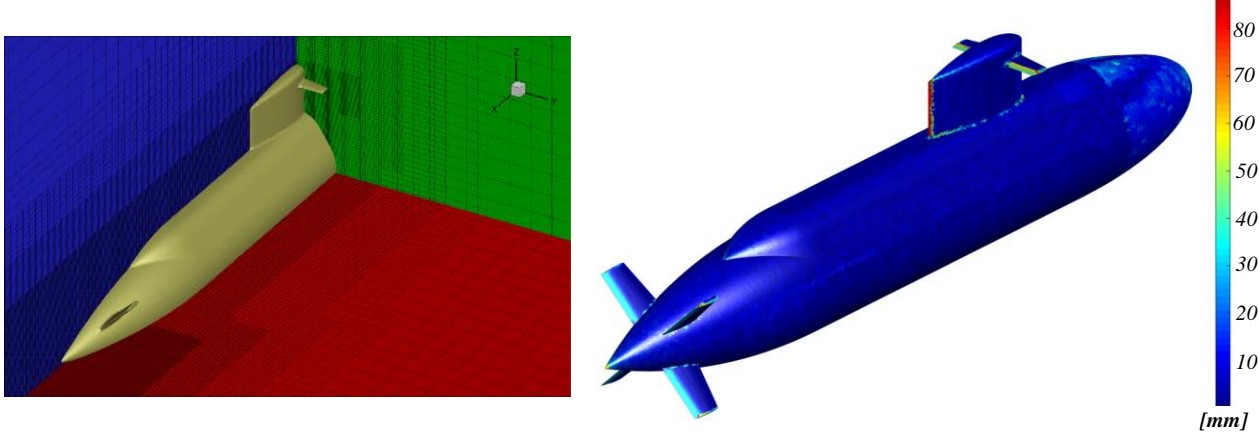

*a) The zero level of the signed distance function with superimposed the grid sections in three perpendicular planes*

*b) The distance map between the original model and the zero level of the signed distance function*

**Figure 8.** The grid generation results.

### 3.2. Mathematical Models and Numerical Algorithms

The CFD simulation was carried out by the immersed boundary algorithm described in [8], to which the reader is referred for details. For the sake of completeness, the key elements of the algorithm are summarized here.

The immersed boundary approach is applied to the solution of the Navier–Stokes equations for incompressible flows. The governing equations that are solved by numerical approximations are here reported with index notation (the repeated index convention is used):

$$\begin{cases} \dfrac{\partial u_i}{\partial x_i} = 0 \\ \dfrac{\partial u_i}{\partial t} + \dfrac{\partial u_i u_j}{\partial x_j} + \dfrac{1}{\rho}\dfrac{\partial p}{\partial x_i} = \dfrac{\partial \tau_{ij}}{\partial x_j} \end{cases} \tag{2}$$

The symbols adopted for physical quantities are:

- $t$ for time;
- $\mathbf{e}_i$, $i = 1, 2, 3$ for the base unit vectors;

- $x_i$, $i = 1, 2, 3$ for spatial coordinates;
- $\mathbf{x} = x_i \mathbf{e}_i$ for the position vector;
- $\rho$ for the density;
- $u_i$, $i = 1, 2, 3$ for the *i*-th velocity component;
- $\mathbf{u} = u_i \mathbf{e}_i$ for the velocity vector;
- $p$ for the pressure;
- $\mu$ for the dynamic viscosity;
- $\nu = \mu / \rho$ for kinematic viscosity;
- $\nu_T$ for the turbulent viscosity;
- and $\tau_{ij} = (\nu + \nu_T)(u_{i,j} + u_{j,i})$ for the stress tensor divided by $\rho$ (the Boussinesq hypothesis was adopted).

Detached Eddy Simulation [11–13] was used for the computation of $\nu_t$ required to model the turbulent stresses. The above equations hold in the fluid domain. On the solid wall, no-slip conditions were applied (i.e., $u_i = 0$), whereas, on the fictitious boundary in the far-field, the velocity was enforced on the inlet boundary, and ambient pressure was fixed on the outlet. As initial conditions, the flow was started from a resting position and accelerated to the final value during a transient of time length given by $L/U_\infty$, $L$ being the body length, and $U_\infty$ the velocity in the far-field.

The equations are discretized by a finite difference approach, where the convective and pressure fluxes are discretized by a fifth-order WENO scheme [14], while the viscous terms are approximated with second-order centered approximation. Time integration was performed by a second-order fully implicit scheme, with a constant time step equal to $U_\infty \Delta t / L = 5 \times 10^{-3}$.

### 3.3. Numerical Set-Up

In all the simulations, the adopted Reynolds number was $Re = 2.7 \times 10^6$, as in the experiments reported in [10].

To enforce the boundary conditions on the submarine walls, the Immersed Boundary procedure described in detail in [8] was applied; the algorithm can be summarized as follows:

1. at the beginning of each time step, the solution is extrapolated inside the body in the normal direction to the body surface;
2. the solution at internal points is then modified in order to get null velocity on the rigid walls;
3. the discrete equations are locally modified to retain at least second order accuracy in the neighborhood of the wall.

Given the high value of the Reynolds number, the wall stresses were evaluated by the use of wall functions, as described in the referenced paper; of course, by this approximation, the details of the boundary layers on the hull are lost, and only the wall stress exerted on the external flow is represented in the model. With the adopted grid, cell size on the walls in terms of wall units is $y^+ = d u_\tau / \nu = O\ (200 \sim 300)$, $d$ being the cell thickness, $u_\tau = \sqrt{\tau_w / \rho}$ the friction velocity, and $\tau_w$ the tangential stress on the wall. Nevertheless, vorticity production on the solid walls and the following evolution in the wakes are very well represented, as shown in the next section.

### 3.4. Results

The computed pressure on the submarine hull is reported in Figure 9, whereas the instantaneous vortex structures are reported by the Q-criterion [15] with $Q = -50$ in Figure 10. From this figure, it can be seen that the grid is able to capture the details of the large vortical structure; in particular, the tip vortices from the sail wings are very well captured, together with their interaction with the vortex structures in the wake of the sail. Similarly, all the details of the large eddies in the wake of the main body and of the tail appendages are captured, and their evolution is very well represented in all the refined regions.

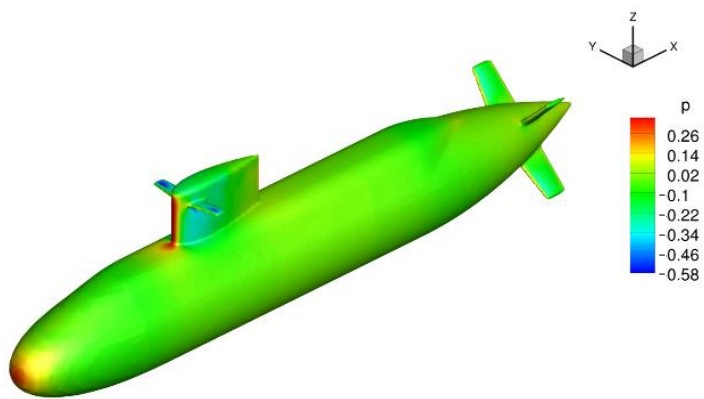

**Figure 9.** Non-dimensional pressure contours of the submarine surface.

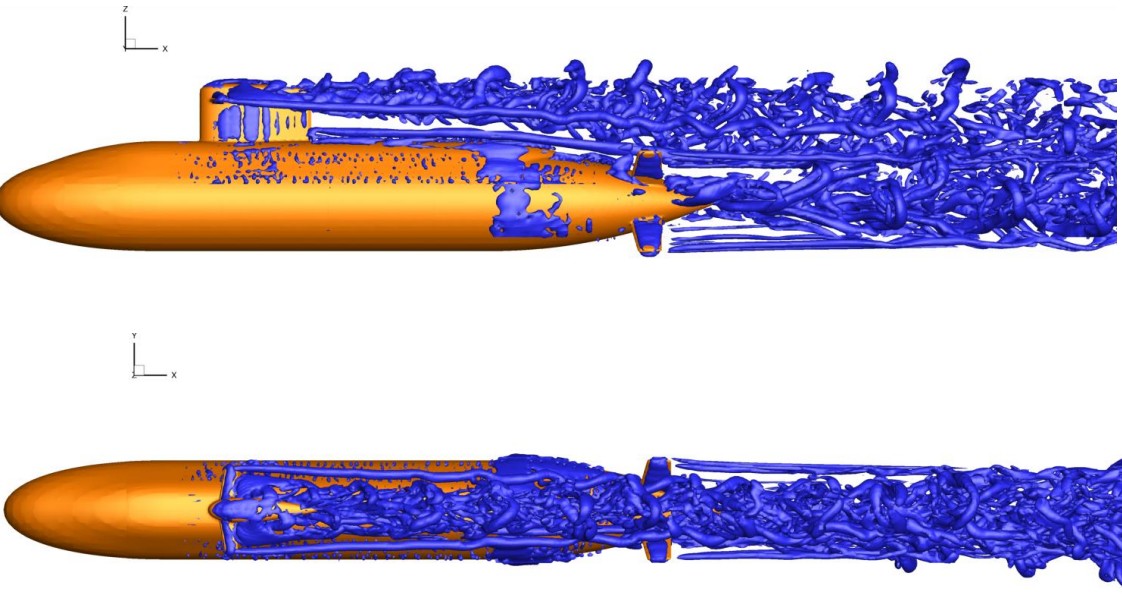

**Figure 10.** Instantaneous vortex structures visualized by the Q criterion (Q = −50).

The time average of the computed solution is reported in the lower part of Figure 11 in terms of axial velocity on the symmetry plane; in the top part of the same figure, the instantaneous contours of the same variable are also reported. In Figure 12, the averaged axial velocity is reported on six cross planes downstream the hull.

The numerical uncertainty was evaluated by following the procedure described in [16], as recommended by most international engineering associations (e.g., International Towing Tank Conference ITTC and American Institute of Aeronautics and Astronautics AIAA). A first level of coarser grid was generated from the finest one by removing every other point in each direction. A third level of coarsening was impossible because the grid would have been too coarse to capture some basic element of both geometry and flow characteristics. Therefore, we adopted the two–grid verification procedure in [16], where the uncertainty $U$ is evaluated as

$$U = F\frac{||u^f - u^c||_1}{r^2 - 1}\frac{1}{|u^f||_1} \times 100 \tag{3}$$

where $r = 2$ is the adopted refinement ratio in each direction, $u^f$ is the solution computed on the fine grid, $u^c$ is the solution on the coarse grid, and $F$ is a safety factor that, as suggested in [16] for the two-grid uncertainty verification, was chosen to be equal to 3. The quantity $||u^f - u^c||_1$ is the $L_1$-norm of the difference between the two solution, whereas $|u^f||_1$ is the $L_1$-norm of the field computed on the fine grid. The uncertainty, computed on the averaged velocity field, was $U = 3.08\%$ for the case considered in the reported example.

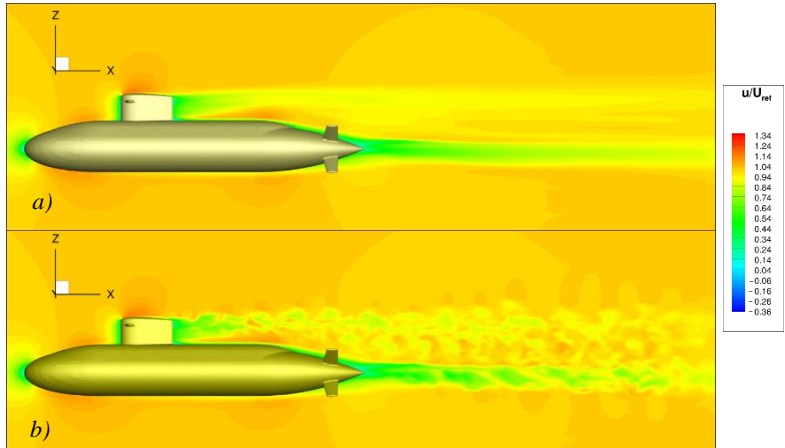

**Figure 11.** Non–dimensional instantaneous (*b*) and averaged (*a*) axial velocity component on the plane $y = 0$.

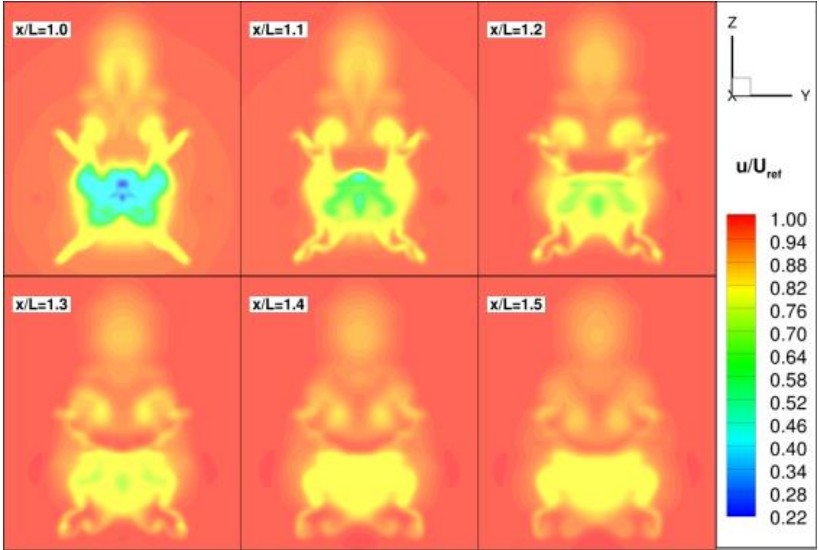

**Figure 12.** Averaged non-dimensional axial velocity on six downstream sections.

Finally, in Figure 13, the contours of the resolved and modeled kinetic energy are reported on both the symmetry plane and on several cross-sections. From this figure, it can be seen that, according to the Pope criterion [17], the grid is adequate for a correct LES simulation, and the ratio between the modeled to the total turbulent kinetic energy being always below 0.2.

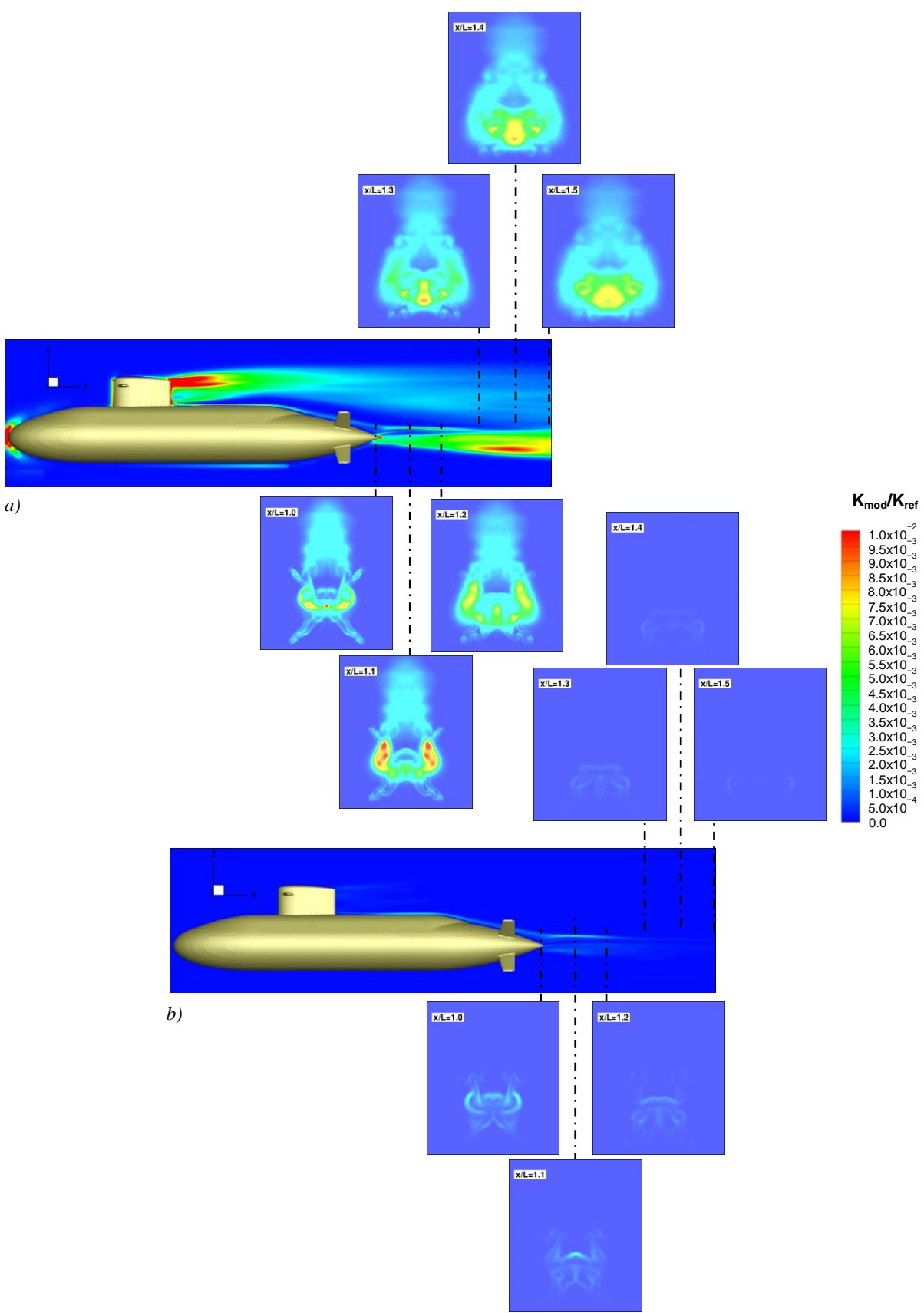

**Figure 13.** Resolved (**a**) and modeled (**b**) turbulent kinetic energy.

## 4. Conclusions

In this paper, an almost fully automatic methodology for CFD analysis for high Reynolds number flows is also presented. The proposed workflow includes a new method for Cartesian Mesh Generation with Local Refinement devised and applied to the IB method developed in [8]. The innovative aspects in the present Cartesian adaptive grid method can be found in the strategies of diversification of the mesh dimensions in the different parts of the model, based on automatic segmentation of the surfaces enveloping the object. In addition, grid refinement can be explicitly controlled in regions where the flow is expected to exhibit high gradients. This, together with the use of the IB method and of the wall functions described in [8], allows the simulation of high Reynolds number flows, with limited grid requirements of the boundary layers. The whole methodology, starting from a discrete manifold model of the object to be analyzed and from the following input:

- definition of the computational domain and regions of interest with the dimensions of the corresponding grid;
- the key information for the CFD simulation (expected high flow gradients);

automatically produce the grid for the CFD analysis. The aim of this paper is to show that this automatic workflow is robust and enables to obtain quantitative results on geometrically complex configurations such as marine vehicles. For this purpose, the proposed methodology was applied to study the flow past a BB2 submarine. The grid was able to capture the details of the large vortical structures from the sail wings and from the tailplanes, as well as their interaction with the wakes emanating from the sail and from the main body. Furthermore, the grid proved to be adequate for a correct LES simulation in the wake.

The present research activity will be extended to include the development of an automated mesh refinement strategy, able to capture flow details without explicit input from the user. Moreover, other operating conditions (underwater maneuvers, surfacing, diving) will be addressed, and the results of the fluid dynamic simulations will be verified and validated against available experimental data. In particular, in future research, the capability of the present Immersed Boundary approach coupled with automated mesh refinement will be checked for free surface flows around surface piercing vessels, like ship hull or submarine vehicles operating at snorkeling depth.

**Author Contributions:** Conceptualization, L.D.A. and A.D.M.; methodology, L.D.A. and A.D.M.; software, L.D.A. and A.D.M.; validation, L.D.A. and A.D.M.; investigation, F.D.; resources, F.D.; data curation, L.D.A. and A.D.M.; writing—original draft preparation, L.D.A., A.D.M., and F.D.; writing—review and editing, A.D.V.; visualization, L.D.A. and A.D.M.; supervision, A.D.M.; project administration, A.D.V. All authors have read and agreed to the published version of the manuscript.

**Funding:** This research received no external funding.

**Institutional Review Board Statement:** Not applicable.

**Informed Consent Statement:** Not applicable.

**Conflicts of Interest:** The authors declare no conflict of interest.

## Abbreviations

The following abbreviations are used in this manuscript:

| | |
|---|---|
| CFD | Computational Fluid Dynamics |
| IB | Immersed Boundary |
| LES | Large Eddy Simulation |
| $\{X_G^M, Y_G^M, Z_G^M\}$ | coordinates of the model centroids |
| ROI | Region of Interest |

| | |
|---|---|
| $\{X_{\min}, Y_{\min}, Z_{\min}\}$ and $\{X_{\max}, Y_{\max}, Z_{\max}\}$ | two extreme vertices of the computational domain |
| $2^{k_G}$ | the computational domain grid dimension |
| $\left\{X_{\min}^{M,i}, Y_{\min}^{M,i}, Z_{\min}^{M,i}\right\}$ and $\left\{X_{\max}^{M,i}, Y_{\max}^{M,i}, Z_{\max}^{M,i}\right\}$ | two extreme vertices of the box containing $i$-th segmented surface |
| $2^{k_{M,i}}$ | the grid dimension of the box containing $i$-th segmented surface |
| $\left\{X_{\min}^{w}, Y_{\min}^{w}, Z_{\min}^{w}\right\}_l$ and $\left\{X_{\max}^{w}, Y_{\max}^{w}, Z_{\max}^{w}\right\}_l$ | two extreme vertices of the $l$-th ROI defined by the operator |
| $2^{k_{w,l}}$ | the grid dimension of the $l$-th box defined by the operator |

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
