# Peer review of "Cartesian Mesh Generation with Local Refinement for Immersed Boundary Approaches"

_jmse, doi:10.3390/jmse9060572_

Round 1

Author Response

In the attached file there are the answers to the reviewer

Reviewer 2 Report

The authors introduced a workflow for the local refined Cartesian with immersed boundary method for high-Reynolds number region. The latest literature is cited and used as a framework for the work presented in this manuscript. The topic is interesting for many CFD solvers. However, there are three main concerns that the authors needs to address:

  • The cascade of boxes of grid refinement: how do the authors keep the consistency of the discretization schemes? One advantage of cartesian grid is that it is easy to find neighbor cells for high-order discretization. With local refinement, how is the consistency considered.

  • No quantified verification or validation: Though the reported work focuses on grid generation, the criterion of a good grid is still to produced validated results. The visual display of the wake vortex structure offers little insights with regard to how accurate is the simulation, and thus how good is the grid.

  • Following comment 2, the conclusion is not really supported by the observation. The conclusion claims that ‘the aim of the paper is …workflow is robust …’ and that ‘the grid proved to be adequate for a correct LES simulation.’ Yes, quantitative results are obtained, but there is no solid verification and validation, and thus there is no measurement as for how good the grid is, how robust it is and how accurate it is. Thus claiming ‘adequate for a correct LES simulation’ is beyond what has been proven

In general, the authors need to provide more details regarding how accurate the generated grid is, i.e., verification and validation, not only results. All CFD provides results that look OK, but accuracy need concrete indicators.

Author Response

(The authors gave the same response as above.)

Round 2

Reviewer 1 Report

Thanks to authors for doing the revision very well.  The paper looks much nicer and I recommend it for publication in the present format. Good luck with your future research.

Reviewer 2 Report

The revision has addressed my main concern on verification and validation and provided with an outlook for future research plan. The manuscript is found acceptable for publication in its current status.